# Small molecule targeting r(UGGAA)$_n$ disrupts RNA foci and alleviates disease phenotype in *Drosophila* model

Tomonori Shibata[1], Konami Nagano[2], Morio Ueyama [3], Kensuke Ninomiya[4], Tetsuro Hirose [4,5], Yoshitaka Nagai [3], Kinya Ishikawa[6], Gota Kawai [2] & Kazuhiko Nakatani [1✉]

Synthetic small molecules modulating RNA structure and function have therapeutic potential for RNA diseases. Here we report our discovery that naphthyridine carbamate dimer (NCD) targets disease-causing r(UGGAA)$_n$ repeat RNAs in spinocerebellar ataxia type 31 (SCA31). Structural analysis of the NCD-UGGAA/UGGAA complex by nuclear magnetic resonance (NMR) spectroscopy clarifies the mode of binding that recognizes four guanines in the UGGAA/UGGAA pentad by hydrogen bonding with four naphthyridine moieties of two NCD molecules. Biological studies show that NCD disrupts naturally occurring RNA foci built on r(UGGAA)$_n$ repeat RNA known as nuclear stress bodies (nSBs) by interfering with RNA–protein interactions resulting in the suppression of nSB-mediated splicing events. Feeding NCD to larvae of the *Drosophila* model of SCA31 alleviates the disease phenotype induced by toxic r(UGGAA)$_n$ repeat RNA. These studies demonstrate that small molecules targeting toxic repeat RNAs are a promising chemical tool for studies on repeat expansion diseases.

[1] Department of Regulatory Bioorganic Chemistry, The Institute of Scientific and Industrial Research (ISIR), Osaka University, Ibaraki, Japan. [2] Department of Life and Environmental Sciences, Faculty of Engineering, Chiba Institute of Technology, Chiba, Japan. [3] Department of Neurotherapeutics, Osaka University Graduate School of Medicine, Suita, Japan. [4] Graduate School of Frontier Biosciences, Osaka University, Suita, Japan. [5] Institute for Genetic Medicine, Hokkaido University, Sapporo, Japan. [6] Center for Personalized Medicine for Healthy Aging, Tokyo Medical and Dental University, Tokyo, Japan.
✉email: nakatani@sanken.osaka-u.ac.jp

Aberrant expansion of specific repeat sequences in the human genome causes >40 neurological disorders called repeat expansion diseases[1-3]. Expansion of >36 CAG repeats in the Huntingtin gene *HTT* and 200 CGG repeats in the *FMR1* gene caused Huntington's disease and Fragile X syndrome, respectively. Long CTG repeats have been identified in the *DMPK* gene of patients of Myotonic Dystrophy type 1 (DM1), although the healthy person has the CTG repeat within the range of 5–38. These diseases have also been classified as trinucleotide repeat diseases because the disease-causing repeat sequences consisted of three nucleotides. Besides these, diseases caused by the aberrant expansion of tetra-, penta-, and hexanucleotide repeats are also reported[4-9].

Recent studies on repeat expansion diseases showed that the transcript of aberrantly expanded repeat, thus, long repeat RNA termed as toxic RNAs would be a cause of most repeat expansion diseases[10-14]. The proposed mechanisms of onset by toxic RNAs are sequestration of RNA-binding proteins (RBPs) in nuclei and unusual translation initiation called repeat-associated non-AUG (RAN) translation[15,16]. RNA foci are the aggregates of toxic RNAs and sequester RBPs such as splicing factors in nuclei, resulting in dysfunction of these proteins. In contrast, RAN translation produces peptides with the repeated amino acid sequence with the tendency of forming aggregates in the cytoplasm. This gain-of-function of toxic RNAs can account for some characteristics of the pathogenesis of repeat expansion diseases.

Repeat DNA and RNA sequences can fold into hairpin structures involving mismatches consisting of partially hydrogen-bonded structures, and these structures are potential binding sites for small molecules. Despite the understandings of potential vital roles of small molecules in treating repeat expansion diseases, the exploration of small molecules targeting toxic RNAs remains a challenge. High-throughput screening methods including fluorescent indicator-displacement assays[17-21], small molecule microarrays[22,23], and phenotypic assays[24,25] have been performed to discover small molecules binding to RNAs. Towards this end, our laboratory has focused on the repositioning of previously developed mismatch-binding ligands (MBLs) that target base mismatches in double-stranded DNA (dsDNA)[26-32]. MBLs are designed to bind to the specific mismatches in dsDNA and do not always bind to the mismatches with the same sequence context in double-stranded RNAs (dsRNAs). We anticipated, however, many chances to discover molecules binding to mismatches in dsRNA by screening from our in-house MBL library. We here report that one of MBLs, naphthyridine carbamate dimer (NCD)[29], binds in high affinity to the 5′-UGGAA-3′/5′-UGGAA-3′ site produced in the hairpin structure of the r(UGGAA)$_n$ repeat RNA causing spinocerebellar ataxia type 31 (SCA31). SCA31 is an autosomal dominant spinocerebellar degenerative disorder, and the patients of SCA31 have 2.5–3.8 kbp multiple pentanucleotide repeats containing d(TGGAA)$_m$, d(TAGAA)$_m$, d(TAAAA)$_n$ and d(TAGAATAAAA)$_n$ in the genome. Among these repeats, d(TGGAA)$_n$ is identified as a pathogenic repeat, as other repeats (i.e., d(TAGAA)$_n$, d(TAAAA)$_n$, and d(TAGAATAAAA)$_n$) are found in unaffected individuals[33]. Nuclear magnetic resonance (NMR) spectra of the NCD-UGGAA/UGGAA pentad complex reveal the characteristic NCD-bound RNA structure involving the formation of hydrogen bonding pair with the naphthyridine and guanine residue, the zigzag binding orientation of two NCD molecules in the pentad site accompanied with adenine flip out, and the *syn* glycosidic conformation for all naphthyridine-bound guanines. In vitro pulldown studies reveal that NCD interfered with the interaction of several RBPs with r(UGGAA)$_n$ repeat RNA. NCD treatment resulted in suppression of RNA foci formation in HeLa cells with overexpressed r(UGGAA)$_n$ SCA31-related RNA and the inhibition of endogenous nuclear stress bodies (nSBs)

assembly upon thermal stress exposure resulting in the disruption of nSB-dependent splicing regulation. Most significantly, feeding NCD to larvae of the *Drosophila* model of SCA31 results in alleviation of disease phenotype induced by toxic r(UGGAA)$_n$ repeat RNA. The observed phenotype changes are most likely due to the interference of the interaction between the r(UGGAA)$_n$ repeat RNA and RBPs by NCD as confirmed in vitro and cells. Studies described here clearly show that small molecules targeting toxic repeat RNAs have enormous potential to alleviate the toxic RNA function in vivo.

Transcription of the d(TGGAA)$_n$ repeat identified as a SCA31-causing repeat[34] produced long r(UGGAA)$_n$ repeat RNAs[35], which formed foci with concomitant co-localization of 43 kDa TAR-DNA binding protein (TDP-43) in addition to >80 protein components[36]. *Drosophila* models of SCA31 expressing long r(UGGAA)$_n$ repeat RNA showed degeneration of the eye morphology, life-shortening, locomotor defects, and exhibited similar pathogenesis observed in human SCA31 such as accumulation of r(UGGAA)$_n$-containing RNA foci. Coincidentally, the r(UGGAA)$_n$ repeat sequence is also found in the HSATIII long noncoding RNAs (lncRNAs), which are transcribed from Satellite III pericentromeric repeat upon thermal stress exposure[37]. HSATIII lncRNAs also sequester multiple RBPs to form RNA foci called nSBs in the nucleus. The formation of nSBs results in the suppression of pre-mRNA splicing (or promote intron retention) of ~500 introns during recovery from thermal stress[38]. Overexpression of motor neuron disease-associated RBPs such as TDP-43 and fused in sarcoma (FUS) in *Drosophila* models of SCA31 mitigated toxicity of r(UGGAA)$_n$ repeat RNA accompanied with the dispersion of RNA foci[36]. These results suggested that binding of these RBPs to r(UGGAA)$_n$ repeat RNA induced the release of the sequestered proteins from the RNA and, more importantly, suggest the possibility of small molecules in reducing the RNA toxicity by their binding to r(UGGAA)$_n$ repeat RNA. This hypothesis is relevant to those proposed for small molecules targeting disease-causing toxic RNAs including r(CUG)$_n$ in DM1[39-41], r(CCUG)$_n$ in DM2[42,43], r(CGG)$_n$ in fragile X-associated tremor/ataxia syndrome[44,45], r(GGGGCC)$_n$ in ALS/FTD[46,47], and r(AUUCU)$_n$ in SCA10[48]. We, therefore, started to screen our in-house chemical library to find molecules binding to r(UGGAA)$_n$ repeat RNA.

## Results

### Screening of small molecules binding to r(UGGAA)$_n$ repeat RNAs from the in-house chemical library.

To explore molecules binding to r(UGGAA)$_n$ repeat RNAs, we have screened our in-house MBL library by surface plasmon resonance (SPR) assay, where three repeat RNAs r(UGGAA)$_9$, r(UAGAA)$_9$, and r(UAAAA)$_9$ were immobilized on its surface. Non-pathogenic r(UAGAA)$_9$ and r(UAAAA)$_9$ repeats were used as control RNAs. Most disease-causing repeat RNAs responsible for RNA-mediated neurodegeneration can form hairpin structures involving internal loops consisting of multiple mismatched base pairs[49]. The r(UGGAA)$_n$ repeat RNA may fold into the hairpin structures, which include the UGGAA/UGGAA pentad containing the 5′-GGA-3′/3′-AGG-5′ internal loop (Fig. 1a). From our previous studies, the guanine-rich internal loop could be good target motifs for MBLs consisting of two *N*-acyl-2-amino-1,8-naphthyridine heterocycles, of which hydrogen bonding surface is complementary to that of guanine base (Fig. 1b). Among 20 compounds we investigate (LC-1~LC-20) (Supplementary Fig. 1), four compounds (LC-5, 8, 9, and 11) showed the marked increase in the SPR signal to the r(UGGAA)$_9$-immobilized surface (Fig. 1c and Supplementary Fig. 2) and the band shift of r(UGGAA)$_9$ by gel electrophoretic mobility shift assay (EMSA) (Fig. 1d) with

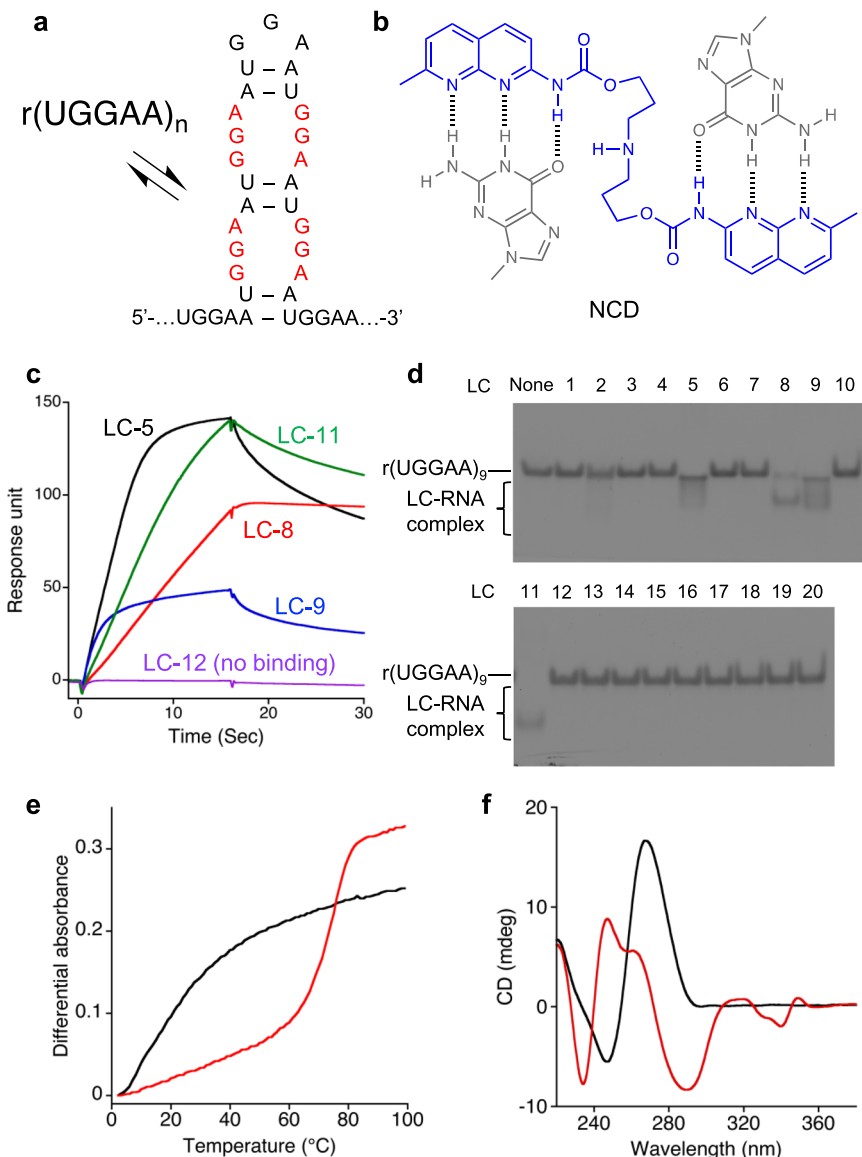

**Fig. 1 Screening of r(UGGAA)$_n$-binding small molecules from an in-house chemical library. a** Possible hairpin structures in r(UGGAA)$_n$ containing internal loop with three consecutive G-A, G-G, and A-G mismatch base pairs (shown in red). **b** Chemical structure of NCD (shown in blue) and schematic illustration of hydrogen bonding to guanines (shown in gray). **c** Representative SPR data for the binding of LC-5 (black), 8 (red), 9 (blue), and 11 (green), and for the non-binding of LC-12 (purple) to r(UGGAA)$_9$-immobilized surface. **d** Native PAGE analysis of interactions between LC-1–20 and r(UGGAA)$_9$. **e** UV melting profiles and **f** CD spectra of r(UGGAA)$_9$ in the absence (black) and presence (red) of NCD. Source data are provided as a Source Data file.

concentration-dependent manner (Supplementary Fig. 3). Structural comparison of four compounds readily suggested that LC-9, which is called NCD showing high affinity to the G-rich DNAs[29,30] would be the minimum structural element for the binding to r(UGGAA)$_9$. The other three compounds were either dimeric form or a very close derivative of NCD.

To gain insights into the NCD binding to r(UGGAA)$_9$, we carried out structure-binding relationship studies on NCD derivatives by EMSA (Supplementary Fig. 4). Neither compounds having different linker structure connecting two naphthyridine heterocycles (LC-2 and 3) nor the molecule having only one naphthyridine heterocycle showed the band shift for the r(UGGAA)$_9$. These results confirmed that NCD is the indispensable structural element for the binding to r(UGGAA)$_9$. UV melting profiles measured for the r(UGGAA)$_9$ in the presence of NCD (20 μM) showed the melting temperature ($T_m$) of 73.7 °C (Fig. 1e), whereas the RNA itself did not show $T_m$. Such a markedly high $T_m$ was not observed for the r(UAGAA)$_9$ in the

presence of NCD, and the r(UAAAA)$_9$ did not show any increase in $T_m$ by NCD (Supplementary Fig. 5). Substantial changes in the circular dichroism (CD) spectra of the r(UGGAA)$_9$ observed upon NCD binding (Fig. 1f) suggested the formation of NCD-bound complex with marked structural changes.

**Sequence selectivity in the binding of NCD to mutated UGGAA/UGGAA pentad.** To know if NCD binding to the r(UGGAA)$_9$ involved the binding to the UGGAA/UGGAA pentad in the secondary hairpin structure, $T_m$s of the RNA duplexes containing the URRAA/URRAA pentad, where R is guanine (G) or adenine (A), were measured in the absence and presence of NCD (Fig. 2a and Supplementary Fig. 6). A marked increase in the $T_m$ by 21.5 °C was observed for the duplex containing the UGGAA/UGGAA pentad (5′-GGA-3′/3′-AGG-5′) in the presence of NCD (red bar), whereas no increase in the $T_m$ was observed with quinoline carbamate dimer (QCD) (blue bar).

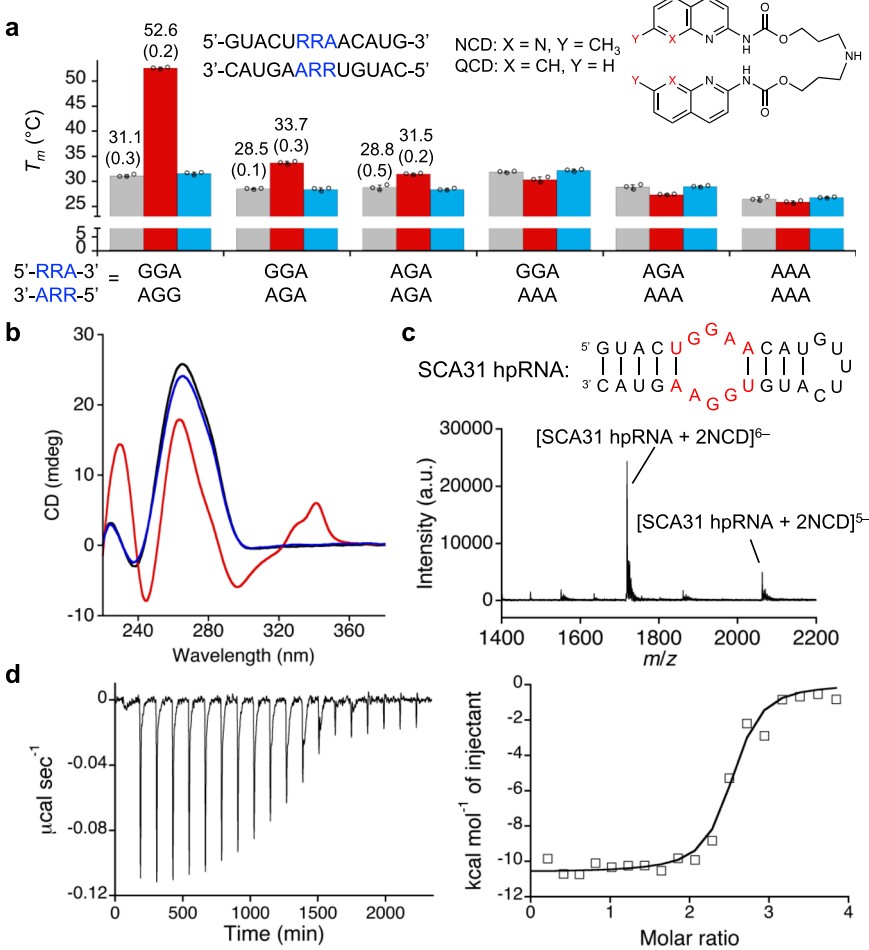

**Fig. 2 Binding of NCD to UGGAA/UGGAA pentad-containing RNA. a** Melting temperatures of RNA duplexes containing URRAA/URRAA pentad in the absence (gray) and presence of NCD (red) and QCD (blue). The data in $T_m$ measurements are presented as the mean ± SD ($n = 3$ independent experiments). **b** CD spectra of RNA duplex containing UGGAA/UGGAA pentad (black) in the presence of NCD (red) and QCD (blue). **c** ESI-TOF-MS spectra of SCA31 hpRNA (10 μM) in the presence of 40 μM NCD. **d** ITC measurements for the binding of NCD to SCA31 hpRNA. Source data are provided as a Source Data file.

QCD did not show the band shift for r(UGGAA)$_9$ in EMSA (Supplementary Fig. 4). Replacing one guanine in 5′-GGA-3′/3′-AGG-5′ with adenine to 5′-GGA-3′/3′-AGA-5′ resulted in a modest $T_m$ increase (5.2 °C) in the presence of NCD. Further substitution of the guanine in 5′-GGA-3′/3′-AGA-5′ with adenine to 5′-AGA-3′/3′-AGA-5′ and 5′-GGA-3′/3′-AAA-5′ diminished the effect of NCD on the increase in $T_m$. Substantial changes on CD spectra of the duplex containing the UGGAA/UGGAA pentad were also observed in the presence of NCD but not QCD (Fig. 2b). These results suggested that NCD binding to the r(UGGAA)$_9$ likely involved the binding to the UGGAA/UGGAA pentad. With the 29-mer hairpin-RNA containing the UGGAA/UGGAA pentad (SCA31 hpRNA; Supplementary Fig. 7), the stoichiometry of NCD binding to the UGGAA/UGGAA pentad was determined to be 2:1 by electrospray ionization time-of-flight mass spectrometry (ESI-TOF MS) analysis (Fig. 2c and Supplementary Fig. 8). The ions corresponding to the 1:1 complex was not detected under the conditions, suggesting that binding of two NCD molecules to the UGGAA/UGGAA pentad would be necessary to stabilize the NCD-SCA31 hpRNA complex. Accurate thermodynamic parameters for the binding of NCD to the UGGAA/UGGAA pentad were determined by isothermal titration calorimetry (ITC) to give $\Delta H$ –10.6 kcal mol$^{-1}$, $\Delta S$ –1.3 cal mol$^{-1}$ K$^{-1}$, and $\Delta G$ –10.2 kcal mol$^{-1}$ (at 25 °C), leading to an apparent dissociation constant $K_{d(app)}$ of 32 nM (Fig. 2d).

**Structural analysis of NCD-UGGAA/UGGAA pentad complex by NMR.** Interactions of NCD with the SCA31 hpRNA (Fig. 3a) were analyzed by NMR spectroscopy. Figure 3b showed the changes in NMR signals of exchangeable protons upon the addition of NCD, indicating the structural change of SCA31 hpRNA. Signals that appeared around 10–12 ppm were due to the formation of four naphthyridine-guanine hydrogen-bonded pairs, which were confirmed by the NOE, as shown in Fig. 3c. It is noted that signals of guanosine residues of the UGGAA/UGGAA pentad could not be observed without NCD, likely due to the exchange broadening, suggesting that the UGGAA/UGGAA region is structurally flexible.

Both exchangeable and non-exchangeable protons of the SCA31 hpRNA-NCD complex were analyzed, and several intermolecular NOEs were found, as shown in Fig. 3c and Supplementary Fig. 9. NMR signals indicated the symmetry in the structure around the UGGAA/UGGAA pentad because the signal of G6 is overlapped with that of G22, and signals of G7 and G23 are close to each other. Signals of four methyl groups in two NCD molecules were observed at 0.7 and around 1.9 ppm, suggesting that two NCD molecules in UGGAA/UGGAA pentad adopt a symmetric structure. The superposition of the ten lowest energy structures was shown in Fig. 3d and the structural statistics were summarized in Supplementary Table 1. Structures of the UGGAA/UGGAA pentad with two NCD molecules were well

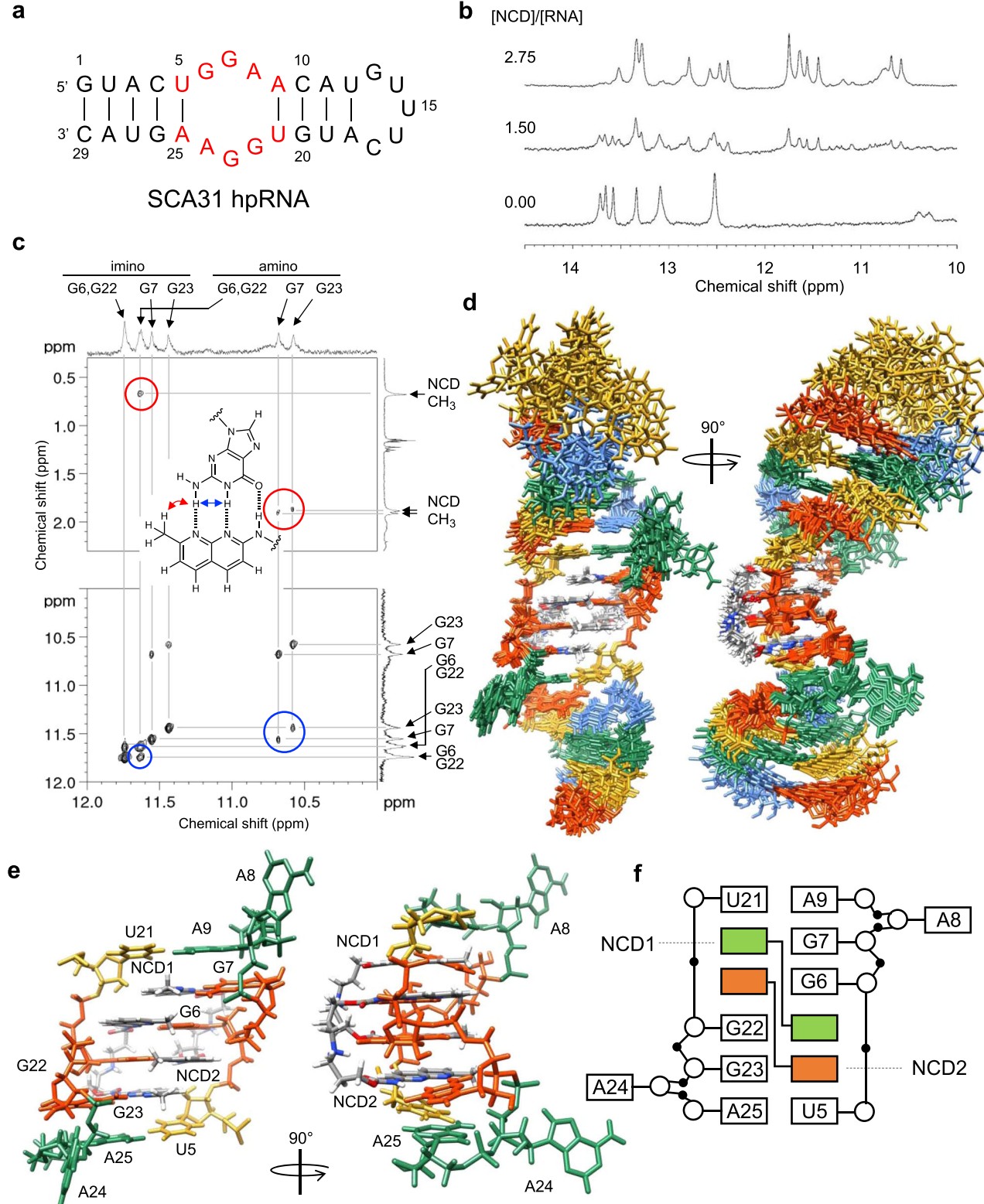

**Fig. 3 NMR structural analysis of NCD-UGGAA/UGGAA pentad complex. a** SCA31 hpRNA used for the NMR analysis. **b** Changes in imino proton spectra of SCA31 hpRNA upon addition of NCD. **c** NOESY spectra of NCD-SCA31 hpRNA complex. Correlations between amino hydrogen and methyl group (red circle) and amino and imino hydrogens (blue circle) were shown with the assignment of hydrogens. **d** Superposition of the ten lowest total energy structures of the NCD-SCA31 hpRNA complex. **e** Close-up structures and **f** structure pattern diagram of the UGGAA/UGGAA pentad bound with two NCD molecules.

defined, and the minimized average structure of the region was shown in Fig. 3e, f. The four guanosine residues bound by naphthyridine were found to be in *syn* conformation regarding the *N*-glycosidic bond; therefore, NCD bound to the UGGAA/ UGGAA pentad from the minor groove. The four naphthyridine-guanine base pairs are stacked between the two A-U base pairs, and the two A residues (A8 and A24) are flipped out toward the major groove. The linker of NCD is located in the minor groove connecting G6 and G23 as well as G22 to G7. It is noted that only the current combination of naphthyridine-guanine pairs provided us acceptable converged structures. Four methyl groups of two NCD molecules are located in line to form a hydrophobic core in the major groove. These structural features of cross-connections and hydrophobic core may contribute to the stabilization of the complex. Two of four methyl groups in NCD were observed in the high field (0.7 ppm), whereas the other two methyl groups were observed at around 1.9 ppm. Naphthyridine rings opposite to G7 and G23 probably caused the ring current shifts observed for the two methyl groups of naphthyridine opposite to G6 and G22, respectively. It is important to note that the methyl group of free NCD was observed at 2.0 ppm. We also analyzed the overall quality of the structure by MolProbity[50] (Supplementary Table 2).

**Inhibitory effect of NCD on the interaction of r(UGGAA)$_n$ repeat RNA with RBPs**. The high-affinity binding of NCD to the r(UGGAA)$_n$ repeat RNA encouraged us to assess the effect of NCD on the interaction between r(UGGAA)$_n$ repeat RNA and RBPs. To investigate the effect of NCD on the RNA–protein interactions, we performed an in vitro pulldown assay using biotinylated r(UGGAA)$_{20}$ with HeLa nuclear extract in the absence and presence of NCD (Fig. 4a). Among RBPs we have tested, TDP-43, heterogeneous nuclear ribonucleoprotein M (HNRNPM), serine and arginine rich splicing factor 9 (SRSF9), and FUS were co-precipitated with the r(UGGAA)$_{20}$, but not with the antisense sequence r(UUCCA)$_{20}$ (Fig. 4b), indicating the sequence-specific binding of these proteins with r(UGGAA)$_{20}$ RNA. In the presence of NCD, we could confirm the inhibitory activity on the co-precipitation of TDP-43, HNRNPM and SRSF9 with the r(UGGAA)$_{20}$ (Fig. 4c, d), but moderate or marginal effects on the co-precipitation of FUS, splicing factor proline and glutamine rich (SFPQ), and ALYREF. These results suggested that NCD could interfere with the interactions between r(UGGAA)$_n$ repeat RNA and TDP-43, HNRNPM, SRSF9, and possibly with other RBPs interacting with SCA31 RNA.

**Inhibition of assembly of RNA foci and nSBs in HeLa cells**. The inhibitory activities of NCD on RNA–protein interactions prompted us to assess the effects of NCD on the assembly of RNA foci and nSBs in HeLa cells. We first analyzed HeLa cells expressing exogenous r(UGGAA)$_{76}$ to examine if NCD disrupts the formation of the RNA foci containing r(UGGAA)$_n$ repeat RNA by fluorescence in situ hybridization (FISH). In the HeLa cells transfected with the plasmid for expressing r(UGGAA)$_{76}$, the accumulation of RNA foci was observed after incubation for 24 h in the nuclei as confirmed by DAPI staining (Fig. 5a; left). In contrast, HeLa cells pre-incubated with NCD showed significantly reduced formation of RNA foci (Fig. 5a, b; middle). The decrease of RNA foci was not observed by pre-incubation with QCD (Fig. 5a, b; right). It is important to note that NCD did not interfere with the interaction of the FISH probe with r(UGGAA)$_n$ repeat RNA, the proliferation of HeLa cells, and the expression of r(UGGAA)$_{76}$ under the conditions (Supplementary Fig. 10). We also observed that NCD interfered with co-localization of TDP-43 with r(UGGAA)$_{76}$, confirming that TDP-43 was released from RNA foci (Supplementary Fig. 11).

HSATIII lncRNAs are induced by thermal stress and sequester various RBPs, including TDP-43, HNRNPM, SRSF9, and FUS, leading to the formation of nSBs[38,51,52]. The effects of NCD on the inhibition of RNA foci assembly raised the intriguing possibility that NCD may inhibit the nSB formation and function in thermal stress-exposed cells by interfering with the sequestration of nSB proteins. We analyzed the effect of NCD on the assembly of nSBs by RNA FISH and immunofluorescence (IF). In the control and QCD-treated HeLa cells, nSBs were normally formed upon thermal stress exposure at 42 °C for 2 h followed by recovery at 37 °C for 1 h and scaffold attachment factor B (SAFB) (Fig. 5c) and SRSF9 (Supplementary Fig. 12) were detected in nSBs. In contrast, in the presence of NCD, nSBs were decreased in a concentration-dependent manner (Fig. 5d, e), and SAFB and SRSF9 were diffused throughout the nucleoplasm (Fig. 5c and Supplementary Fig. 12). As compared with nSBs, the NCD treatment gave only a weak effect on another nuclear body, such as nuclear speckle, including SRSF2 (Fig. 5d, e). NCD may weakly bind to the G-rich SRSF2-binding site of GGNG to induce weak effect on nuclear speckle. The reverse transcription-quantitative PCR (RT-qPCR) analysis confirmed that the amount of HSATIII lncRNAs in NCD-treated cells is comparable with those in the control cells (Fig. 5f), suggesting the disruption of nSBs is not due to the degradation of HSATIII. These observations suggested that NCD interfered with the assembly of the r(UGGAA)$_n$ repeat RNA-associated RNA foci and nSBs in HeLa cells.

**Effect of NCD on nSB-dependent intron retention**. nSBs serve as a part of auto-regulation mechanisms of phosphorylation of SRSFs and splicing of *CLK1* pre-mRNA during thermal stress and subsequent stress recovery (Fig. 6a)[38]. Introns 3 and 4 of *CLK1* pre-mRNA are retained in normal conditions (e.g., 37 °C) and excised upon thermal stress exposure (e.g., 42 °C), which is promoted by thermal stress-dependent de-phosphorylation of SRSFs. During stress recovery, the intron-retaining *CLK1* pre-mRNA is re-accumulated through CLK1-dependent re-phosphorylation of SRSFs. The re-accumulation of the intron-retaining *CLK1* pre-mRNA is accelerated through the efficient re-phosphorylation within nSBs[38,53,54]. As shown in Fig. 6b, c, the intron-retaining *CLK1* pre-mRNA was detected in normal conditions (lanes 1, 4 and 7) and converted to intron-excised mRNA upon thermal stress (lanes 2, 5 and 8) for the control and NCD- or QCD-treated cells. In contrast, re-accumulation of intron-retaining *CLK1* pre-mRNA was significantly suppressed only in the NCD-treated cells (lane 6, *cf.* lanes 3 and 9), suggesting that NCD specifically inhibits the nSB-dependent part of the auto-regulation mechanisms of *CLK1* pre-mRNA splicing. Consistently, NCD showed a limited effect on thermal stress-responsive alternative splicing of pre-mRNAs of *HSP105* and *TNRC6a* (Fig. 6b), which are unaffected by nSBs.

**Effect of NCD on disease phenotype in *the Drosophila* model of SCA31**. Having demonstrated that NCD disrupted r(UGGAA)$_n$ RNA foci by interfering with RNA–protein interaction, we next investigated the biological activity of NCD in the *Drosophila* model of SCA31[36]. This *Drosophila* model of SCA31 expresses pathogenic r(UGGAA)$_{80–100}$ repeats (r(UGGAA)$_{exp}$) in the compound eyes and showed degenerated eye morphology with depigmentation and decrease of the eye area (Fig. 6d; upper panel). *Drosophila* that expresses non-pathogenic repeats including r(UAGAA)$_n$ and r(UAAAAUAGAA)$_n$ was used as a control (Fig. 6d; lower panel, r(UAGAA)(UAAAAUAGAA)$_{exp}$). When NCD (100 μM) was fed to larvae of the *Drosophila* model of SCA31, the degeneration of compound eyes in adults was suppressed (Fig. 6d; upper panel, middle image). The changes in

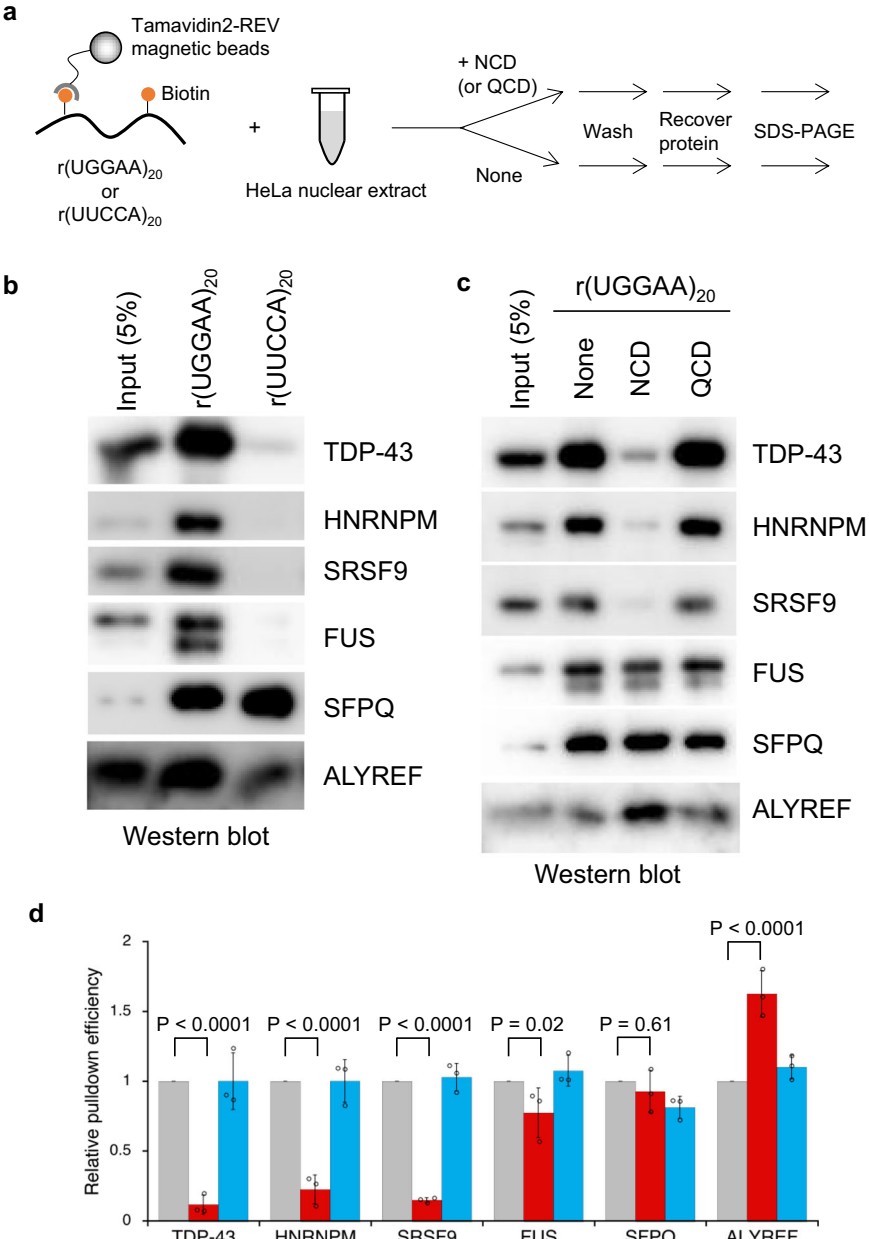

**Fig. 4 Inhibition of interaction between r(UGGAA)_n and RBPs by NCD. a** Workflow of in vitro pulldown assay. **b** In vitro pulldown assay of RBPs using biotinylated r(UGGAA)_{20} or antisense r(UUCCA)_{20}. The co-precipitated proteins were detected by western blotting using specific antibodies. **c** In vitro pulldown assay using biotinylated r(UGGAA)_{20} in the absence and presence of 2 μM NCD and QCD. **d** Relative pulldown efficiencies of each protein in the absence (gray) and presence of NCD (red) and QCD (blue). Data are shown as the mean ± SD ($n = 3$ independent experiments); (Dunnett's multiple comparison test, two-sided). Source data are provided as a Source Data file.

the relative eye area between the NCD-treated and non-treated *Drosophila* expressing r(UGGAA)_{exp} were statistically significant (Fig. 6e; left). In marked contrast, QCD did not show any significant changes both in the morphology of compounds eye (Fig. 6d; upper panel, right image) and the eye area (Fig. 6e; left). For the control, *Drosophila* expressing r(UAGAA)(UAAAAUA-GAA)_{exp} was affected neither by NCD nor QCD at all regarding morphology and eye area of compound eyes (Fig. 6d lower panel, and Fig. 6e right).

## Discussion

As the roles of functional noncoding RNAs (ncRNAs) in biological phenomenon and diseases became apparent, much needs for

the small molecules binding to biologically relevant RNAs appeared. Synthetic RNA-binding molecules increase their importance in consideration of a growing number of evidence that various functional ncRNAs are involved in biological phenomenon and diseases. Among diverse approaches to discover RNA-binding molecules, our group has taken a repositioning approach of MBLs. MBLs consisted of two heterocycles having hydrogen bonding groups located at the edge of the heterocyclic ring, and the heterocycle can form the hydrogen-bonded pair with the specific nucleotide base. The hydrogen-bonded pairs would be further stabilized by stacking with the neighboring base pairs within the interior of the DNA π-stack[26]. The molecule NCD we described here showed $K_{d(app)}$ of 180 nM to the G-G

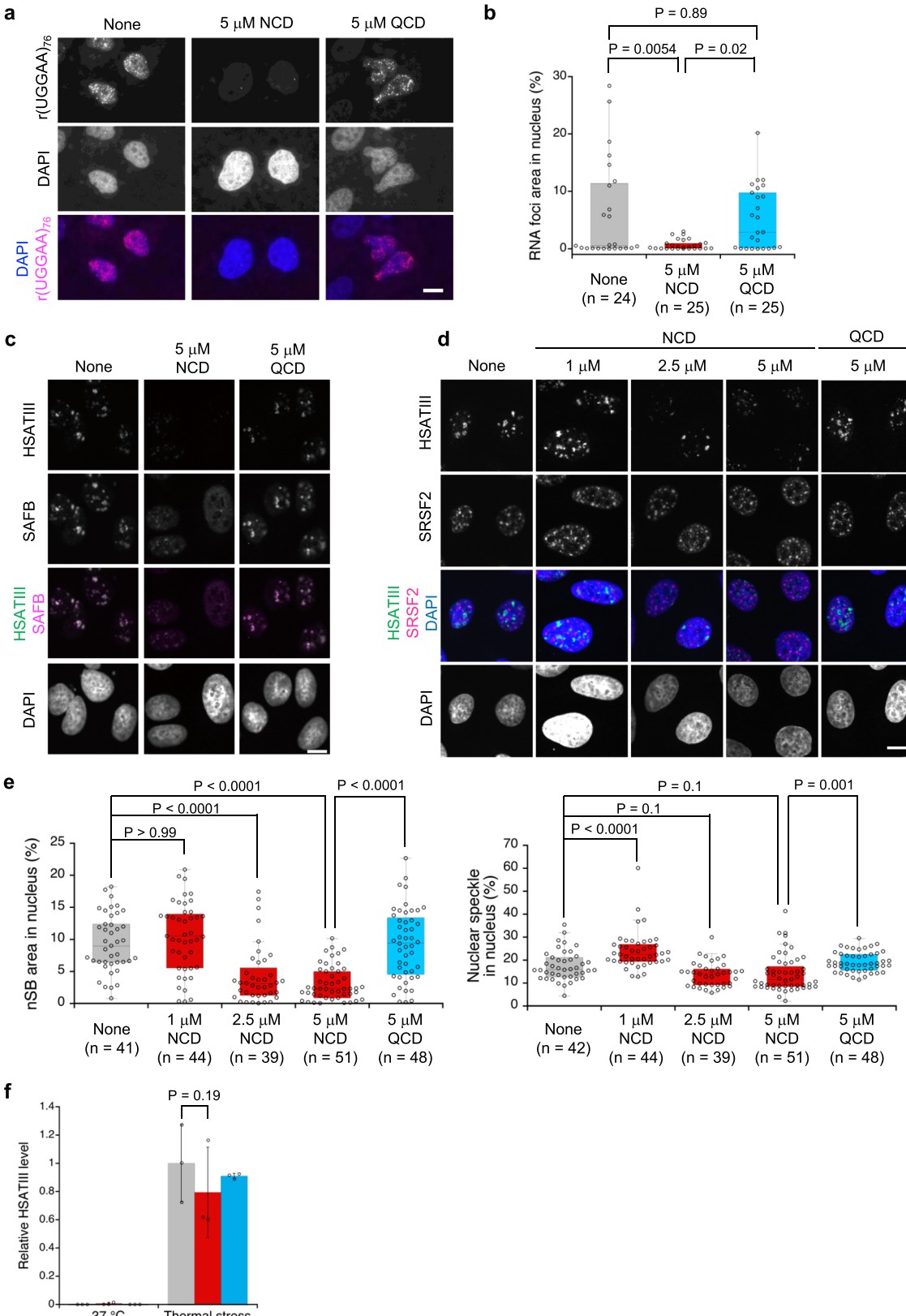

mismatched base pair in dsDNA with the sequence context of 5′-CGG-3′/3′-GGC-5′ (Supplementary Fig. 13), In contrast, $K_{d(app)}$ for the NCD binding to UGGAA/UGGAA pentad in dsRNA was 32 nM (Fig. 2d), which was six-fold lower as compared with $K_{d(app)}$ for the binding to 5′-CGG-3′/3′-GGC-5′ in dsDNA.

One of the common structural characteristics observed for the toxic RNAs is the formation of hairpin structures involving multiple mismatched base pairs[49]. The expanded r(UGGAA)$_n$ repeat RNA, causing SCA31 likely takes various secondary and tertiary structures, and these structures are under the dynamic

**Fig. 5 The effect of NCD on RNA foci and nSB assembly. a** RNA FISH images of cells expressing r(UGGAA)$_{76}$ in the absence (left) and presence of NCD (middle) and QCD (right). Ligand concentration was 5 μM. Scale bar, 10 μm. ($n = 3$, independent experiments). **b** Box plots of total area of r(UGGAA)$_{76}$ RNA foci in each nucleus ($n$ = the analyzed cell number). $p$-value (Kruskal–Wallis test, followed by Dunn's multiple comparison test, two-sided) was shown above. **c, d** RNA FISH and IF images of HeLa cells after thermal stress exposure in the absence and presence of NCD (1, 2.5, and 5 μM) or QCD (5 μM) stained by HSATIII-FISH and IF using an anti-SAFB (**c**) and anti-SRSF2 (**d**) antibodies. Scale bar: 10 μm. ($n = 3$ for **c** and 2 for **d**, independent experiments). **e** Box plots of total area of nSBs (left) and nuclear speckles (right) in each nucleus ($n$ = the analyzed cell number). $p$-value (Kruskal–Wallis test, followed by Dunn's multiple comparison test, two-sided) was shown above. **f** RT-qPCR analysis of HSATIII RNA level. HeLa cells were cultured at 37 °C for 3 h (left; 37 °C) or at 42 °C for 2 h followed by recovery at 37 °C for 1 h (right; thermal stress) in the absence (gray) and presence of 5 μM NCD (red) or QCD (blue). Data are shown as the mean ± SD ($n = 3$ independent experiments); (Dunnett's multiple comparison test, two-sided). Box plots in **b** and **e** show median (center line), upper and lower quartiles (upper and lower box limits), upper quartiles +1.5× interquartile range (upper whiskers), and lower quartiles −1.5× interquartile range (lower whiskers). Source data are provided as a Source Data file.

equilibrium. We made use of our in-house MBL library containing heterocycles with different hydrogen bonding surface and identified NCD as the hit compound binding to the r(UGGAA)$_n$ repeat RNA. The binding site of NCD in r(UGGAA)$_n$ repeat RNA was firmly determined to be the UGGAA/UGGAA pentad in the secondary hairpin structure by NMR structural analysis, which also showed the substantial structural changes induced on the RNA. In the NCD-SCA31 RNA complex, NCD bound to the RNA from the minor groove side. Each one of four guanines in the UGGAA/UGGAA pentad formed the hydrogen bonds to the *N*-acyl-2-amino-1,8-naphthyridine in NCD, and the produced guanine-naphthyridine hydrogen-bonded pairs were stacked inside the π-stack in dsRNA. Remarkably, all four guanosine residues bound by naphthyridine formed a *syn* conformation regarding the glycosidic bond. Previously, we reported the NMR structure of the complex of CAG/CAG triad DNA with two molecules of naphthyridine-azaquinolone (NA), where the guanine-naphthyridine and adenine-azaquinolone hydrogen-bonded pairs were produced[28]. In the case of the NA-CAG/CAG triad complex, NA bound to DNA from the major groove side and the glycosidic bond in guanosine and adenosine bound by NA adopt the *anti* conformation.

Our in vitro pulldown assay showed that NCD inhibited RNA–protein interaction between r(UGGAA)$_n$ repeat RNAs and the RBPs, including TDP-43, HNRNPM, and SRSF9 that preferentially bind to the r(UGGAA)$_n$ repeat sequence. Previous studies reported that GAAUG was identified as a TDP-43-binding motif[55]. Furthermore, it has been reported that HNRNPM and SRSF9 preferentially bound to purine-rich sequences[55,56]. Our NMR structural analysis of the NCD-UGGAA/UGGAA complex indicated the NCD binding to 5′-GGA-3′/3′-AGG-5′ internal loop through sequestration of four guanines. Notably, the recognition sequence of NCD in r(UGGAA)$_n$ is overlapped with those of TDP-43, HNRNPM, and SRSF9, supporting the inhibitory effect of NCD on the specific interaction of r(UGGAA)$_n$ with these RBPs. While a moderate decrease in pulldown efficiency was observed in FUS, it is likely due to multiple RNA-binding modes of FUS by sequence and shape recognition through two separate domains[57].

We further investigated the mode of action of NCD to r(UGGAA)$_n$ using the thermal stress-induced nSBs consisting of HSATIII lncRNAs as a model system, demonstrating that NCD impaired the nSB assembly without affecting HSATIII lncRNA levels. It has been shown that the RBP(s) harboring intrinsically disordered regions such as FUS and TDP-43 play a crucial role in the assembly of specific RNA foci[58,59]. Therefore, the effect of NCD was attributed to impairment of RNA–protein interactions required for the nSB assembly that consequently leads to suppression of the nSB-dependent intron retention that we recently reported[38].

Our studies on biological activity using the SCA31 models demonstrated reduction of the RNA foci in r(UGGAA)$_n$-expressing cells and alleviation of the disease phenotype in the

*Drosophila* model of SCA31. It is remarkably noticeable that the difference in the effect of NCD and QCD on the SCA31 disease phenotype strongly correlates with their differences in the results of our in vitro binding, pulldown and cell-based assays. Although the elucidation of precise mechanisms alleviating the disease phenotype by NCD needs further studies, the disruption of r(UGGAA)$_n$ RNA foci in the SCA31 cell model by NCD likely occurs in a similar mode of action to that of nSBs. Preceding studies demonstrated that accumulation of r(UGGAA)$_n$ RNA foci clearly correlated with the severity of disease phenotype in *Drosophila* model of SCA31[36]. In addition, the disruption of the RNA foci by binding of TDP-43 to r(UGGAA)$_n$ alleviated the disease phenotype in *Drosophila* model of SCA31[36]. The high affinity and specific binding of NCD to SCA31 repeat RNA may prevent pathogenic functions in r(UGGAA)$_n$ such as RNA misfolding and RNA–protein interaction, eventually leading to the rescue of phenotype in *Drosophila* model of SCA31. We cannot exclude concerns regarding the off-target effect of NCD on other G-rich sequences existing in the human genome and transcripts because NCD was originally designed to bind to the G-rich sequences. However, the phenotypic change in *Drosophila* model of SCA31 by NCD is most likely due to the NCD binding to r(UGGAA)$_n$, because of high binding affinity of NCD to the UGGAA/UGGAA pentad, specific suppression of nSB-dependent splicing event by NCD, and the structure-activity relationship between NCD and QCD in the phenotypic change of *Drosophila* model of SCA31. Our demonstration of the mode of NCD-action to nSBs and the phenotype changes in the *Drosophila* model of SCA31 by NCD provides a potential therapeutic strategy of SCA31 for targeting r(UGGAA)$_n$ RNAs by small molecules. The well-defined structures of NCD-RNA complex and bioactivity of NCD for the SCA31 disease model suggest that targeting mismatched base pairs in toxic repeat RNAs by small molecules could be a potential therapeutic strategy to treat repeat expansion diseases.

## Methods

**Surface plasmon resonance (SPR) assay**. 5′-biotin-TEG r(UGGAA)$_9$, r(UAGAA)$_9$ and r(UAAAA)$_9$ repeat RNAs were immobilized on the SA sensor chip (BIAcore) that coated the surface with streptavidin. The surface of sensor chip SA was washed with 50 mM NaOH and 1 M NaCl at three times for 60 s with the flow rate of 30 μl min$^{-1}$. 5′-biotin-TEG repeat RNAs were immobilized to the surface under the following conditions: 200 nM repeat RNA in 10 mM HEPES (pH 7.4), 500 mM NaCl. Amount of r(UGGAA)$_9$, r(UAGAA)$_9$ and r(UAAAA)$_9$ immobilized on the chip surface was 586, 658, and 529 response units (RU), respectively. SPR analysis for the binding of in-house chemical library to the repeat RNA-immobilized surfaces was performed using a BIAcore T200 SPR system (GE Healthcare) under the following condition: 500 nM compounds in HBS-EP + buffer (GE Healthcare) containing 10 mM HEPES (pH 7.4), 150 mM NaCl, 3 mM EDTA, 0.05% v/v Surfactant P20.

**Electrophoretic mobility shift assay (EMSA)**. Repeat RNA (200 nM) without and with ligand (2 μM) in 10 mM sodium cacodylate buffer (pH 7.0) containing 100 mM NaCl was incubated at room temperature for 15 min. The mixtures were mixed with 10× loading buffer (TAKARA) and were subjected to electrophoresis through a native polyacrylamide gel with 1× Tris/Borate/EDTA buffer at room

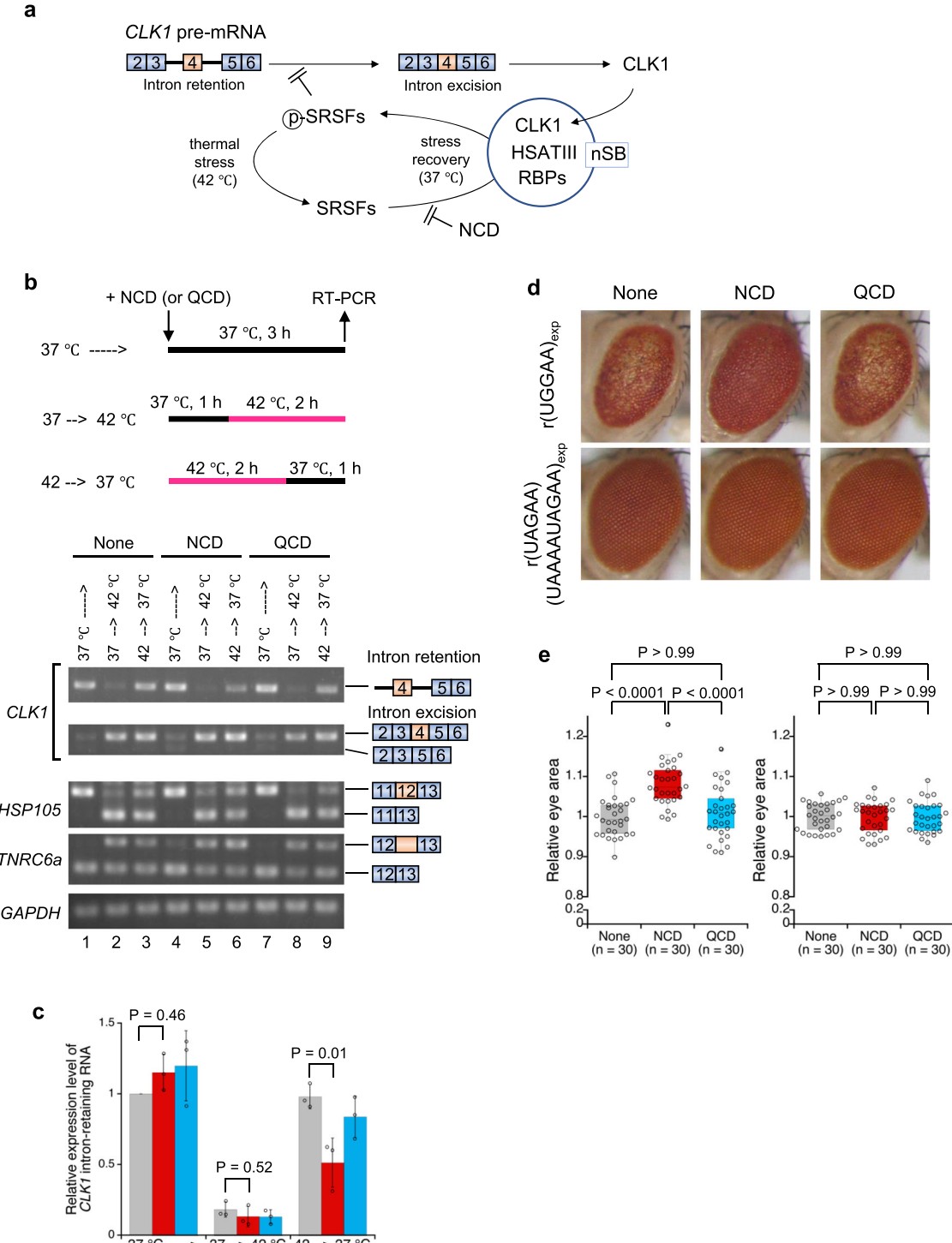

**Fig. 6 Biological activity of NCD in splicing and SCA31 disease models. a** Schematic illustration of nSB-dependent intron retention through re-phosphorylation of SRSFs. **b** Semi-quantitative RT-PCR analysis of *CLK1* intron-retaining pre-mRNA during thermal stress recovery. The *GAPDH* mRNA was used as an internal control. **c** Relative expression level of the *CLK1* intron-retaining pre-mRNA in the absence (gray) and presence of 2.5 μM NCD (red) and QCD (blue). Data are shown as the mean ± SD ($n = 3$ independent experiments); (Dunnett's multiple comparison test, two-sided). **d** Light microscopic images of compound eyes from *Drosophila* expressing r(UGGAA)$_{exp}$ (*GMR-Gal4/+; UAS-*(*UGGAA*)$_{exp}$/+) (*upper, left*) and r(UAGAA)(UAAAAUAGAA)$_{exp}$ (*GMR-Gal4/+; UAS-*(*UAGAA*)(*UAAAAUAGAA*)$_{exp}$/+) (*lower, left*) with the administration of NCD (middle) and QCD (right). Ligand concentration was 100 μM. **e** Box plots of the eye area of *Drosophila* expressing r(UGGAA)$_{exp}$ (left) and r(UAGAA)(UAAAAUAGAA)$_{exp}$ (right) ($n$ = the number of the analyzed flies). Box plots show median (center line), upper, and lower quartiles (upper and lower box limits), upper quartiles +1.5× interquartile range (upper whiskers), and lower quartiles –1.5× interquartile range (lower whiskers). *p*-value (Kruskal–Wallis test, followed by Dunn's multiple comparison test, two-sided) was shown above. Source data are provided as a Source Data file.

temperature. After electrophoresis, the gels were stained for 10 min with SYBR Gold (Thermo).

**Melting temperature measurements**. Thermal denaturation profiles were recorded on a UV-2700 spectrophotometer (Shimadzu) equipped with the TMSPC-8 temperature controller. The absorbance of RNAs (4 μM for RNA duplexes and 2 μM for repeat RNAs) without and with ligand (20 μM) in 10 mM sodium cacodylate buffer (pH 7.0) containing 100 mM NaCl was monitored at 260 nm from 2 to 100 °C (1 °C min⁻¹). $T_m$ was calculated using the median method.

**ESI-TOS-MS measurements**. Samples were prepared by mixing UGGAA/UGGAA pentad-containing hairpin-RNA (10 μM) and NCD (5–40 μM) in water containing 100 mM ammonium acetate and 50% methanol. Mass spectra were obtained with JEOL JMS-T100LP AccuTOF LC-plus 4 G mass spectrometer in negative mode. Spray temperature was fixed at –10 °C.

**CD measurements**. CD experiments were carried out on a J-725 CD spectrometer (JASCO) using a 10 mm path length cell. CD spectra of RNAs (4 μM for RNA duplex and 2 μM for repeat RNA) in the absence and presence of ligand (20 μM) were measured in 10 mM sodium cacodylate buffer (pH 7.0) containing 100 mM NaCl.

**ITC measurements**. A solution of UGGAA/UGGAA pentad-containing hairpin-RNA (2.5 μM) or CGG/CGG-containing hairpin DNA (4 μM) was titrated with NCD solution (50 μM for RNA or 80 μM for DNA) at 25 °C in 10 mM sodium cacodylate buffer (pH 7.0) containing 100 mM NaCl on a MicroCal iTC$_{200}$ calorimeter. Thermodynamic parameters were calculated from the binding curve using Microcal origin 7.0 with a binding model involving a single set of identical sites.

**NMR structural analysis**. RNA sample for NMR measurements was purchased from GeneDesign. RNA sample was dissolved in 20 mM sodium phosphate buffer (pH 6.9) with 100 mM NaCl and 5% D$_2$O. The sample concentration was 75 μM. NMR spectra were measured at 288 K with Avance600 spectrometer (Bruker BioSpin). For water signal suppression, the jump-and-return pulse[60] was used for the 1D imino proton spectra, and the 3-9-19 pulse[61] were used for other measurements. After the NMR measurement in free form, the SCA31 hpRNA was titrated by NCD with molar ratio of 1:1.5, 1:2, 1:2.5, 1:2.75, 1:3, and 1:3.5. Then, the excess amount of NCD was removed by ultrafiltration by Vivaspin (Amicon). The solvent was replaced by 100% D$_2$O, and NMR spectra were measured for structural determination.

NMR data were processed with TopSpin (Bruker Biospin) and analyzed with Sparky[62]. Most of the signals for base protons and H1' protons were assigned for RNA, and all CH protons of NCD were assigned. Structure of the RNA-NCD complex was calculated with CNS_SOLVE[63]. One-hundred ninety-two distance restraints including 36 hydrogen bonding restraints, 157 dihedral restraints, and 14 planarity restraints were used (Supplementary Table 1). The geometry and atomic charge for NCD in an extended conformation was prepared with UCSF Chimera[64] and ANTECHAMBER[65].

Forty-four structures without any violation for restraints were obtained by 100 calculations. Ten structures with the lowest energy were chosen, and the averaged-minimized structure was obtained. R.m.s.d. for the ten structures for heavy atoms was 1.966 ± 0.775 and 1.069 ± 0.388 Å for all and the core region, U5-G7, A9, U21-G23, A25, and NCD, respectively.

**Plasmid construction**. Preparation of plasmid containing d(TGGAA)$_{76}$ was delegated to Thermo Fisher Scientific. The d(TGGAA)$_{76}$-containing plasmid was digested with *Eco*RI and *Xba*I, and inserted into *Eco*RI–*Xba*I site of pCMVTnT to give pCMVTnT-d(TGGAA)$_{76}$ plasmid.

**Cell culture**. HeLa cells (originally obtained from RIKEN BRC) were cultured at 37 °C in 5% CO$_2$ in Dulbecco's modified eagle's medium (Sigma) containing 10% fetal bovine serum (MP Biomedicals), penicillin and streptomycin (Thermo). For thermal stress induction, the cells were incubated at 42 °C in an incubator with 5% CO$_2$.

**In vitro pulldown assay**. In vitro pulldown assay was performed as previously reported[66]. In brief, d(GGAAT)$_{20}$ dsDNA was introduced into pCR-Blunt II-TOPO vector (Thermo Fisher). Then, UGGAA and the antisense repeat RNAs were synthesized by in vitro transcription using T7 and SP6 RNA polymerases, respectively with Biotin RNA labeling mix (Sigma) and purified using Centri-Sep Spin Columns (Thermo Fisher). Biotinylated r(UGGAA)$_{20}$ or r(UUCCA)$_{20}$ repeat RNA (1 μg) was bound to Tamavidin2-REV magnetic beads, mixed with pre-absorbed HeLa nuclear extract diluted in PBS containing 0.1% Triton X-100, 1× protease inhibitor, 1 mM PMSF and RNase inhibitor without and with NCD or QCD (2 μM), and rotated overnight at 4 °C. After washing five times with cold PBS

containing 0.1% Triton X-100, the co-precipitated proteins were eluted in SDS sample buffer for 5 min at 95 °C.

**Western blotting**. Protein extracts in 1× SDS sample buffer were boiled for 5 min, separated by sodium dodecyl sulfate polyacrylamide gel electrophoresis, and transferred to a polyvinylidene fluoride membrane (Millipore) by electroblotting. After incubation with primary and HRP-conjugated secondary antibodies, the signals on the membranes were developed by a chemiluminescence reaction using the ImmunoStar Kit (Wako Chemicals), detected with a ChemiDoc imaging system (BioRad), and analyzed using ImageJ software (NIH). Antibodies are as follows. Anti-HNRNPM (LifeSpan Biosciences, LS-B2427), anti-SFPQ (MBL, RN014MW), anti-SRSF9 (MBL, RN081PW), anti-ALYREF (Santa cruz, sc323-11), anti-TDP-43 (Proteintech, 10782-2-AP), anti-FUS (Santa cruz, sc477-11), anti-Mouse IgG (HRP-Linked) (GE Healthcare, NA931-1ML), anti-Rabbit IgG (HRP-Linked) (GE Healthcare, NA934-1ML).

**RNA fluorescence in situ hybridization (FISH) and immunofluorescence (IF)**. A total of $1 \times 10^5$ cells were seeded into 24-well plates containing coverslips. After a 24 h culture, the cell culture medium containing compound was added to the cells. Plasmids (500 ng) expressing r(UGGAA)$_{76}$ were transfected to the cells using 2 μl of FuGENE HD Transfection Reagent (Promega). After 24 h of transfection, the cells were washed with phosphate-buffered saline (PBS) and fixed with 4% paraformaldehyde for 30 min at 4 °C. The fixed cells were washed with PBS and permeabilized with PBS containing 2% acetone pre-chilled at –20 °C for 5 min at 4 °C. Subsequently, the cells were washed with PBS and stored in 70% EtOH at –20 °C overnight. After washing with PBS and rehydration with 30% formamide in 2× saline sodium citrate (SSC) for 10 min at room temperature, the cells were pre-hybridized in hybridization buffer (30% formamide, 2× SSC, 66 μg ml⁻¹ yeast tRNA, 0.02% BSA, 10% dextran sulfate, 2 mM Ribonucleoside-Vanadyl Complex) for 30 min at 37 °C and hybridized in hybridization buffer containing 1 nM Alexa647-labeled (TTCCA)$_5$ DNA/LNA probe (sequence detail was shown in Supplementary Fig. 10) for 2 h at 37 °C. The washing of the coverslips was performed three times in 2× SSC containing 50% formamide, two times in 1× SSC and two times in 0.1× SSC for 20 min at 55 °C. The cells were mounted onto microscope slides with SlowFade Diamond containing DAPI (Thermo). Fluorescence images of cells were taken by BZ-9000 Fluorescence Microscope (KEYENCE). FISH and IF staining of nSBs and nuclear speckles were performed as previously reported[38]. Antibodies used as are follows (also see western blotting section). Anti-SAFB (Abcam, ab8060), anti-SRSF2 (Sigma-Aldrich, S4045), anti-Digitonin (Abcam, ab420 and ab76907), anti-TDP-43 (Proteintech, 10782-2-AP), anti-Mouse IgG (H + L) (Alexa Fluor488) (Thermo Fisher Scientific, A11029), anti-Rabbit IgG (H + L) (Alexa Fluor488) (Thermo Fisher Scientific, A11034), anti-Rabbit IgG (H + L) (Alexa Fluor568) (Thermo Fisher Scientific, A11036), anti-Goat IgG H&L (Alexa Flour 488) (Abcam, ab150129), anti-Mouse IgG H&L (Alexa Fluor568) (Abcam, ab175472). The image analysis was performed using ImageJ (NIH) software. Areas of r(UGGAA)$_{76}$ RNA foci, nSBs, nuclear speckles, and nuclei were defined and measured by binarized images of r(UGGAA)$_{76}$ RNA, HSATIII, SRSF2, and DAPI, respectively.

**Reverse transcription-quantitative PCR (RT-qPCR)**. RT-qPCR and semi-quantitative RT-PCR were performed as previously reported[38]. In brief, total RNAs were prepared using TRI Reagent (Molecular Research Center, Inc.), according to the manufacturer's manual. The RNAs were treated with RQ1 RNase-free DNase (Promega) according to the manufacturer's manual. First-strand cDNA was synthesized using High-Capacity cDNA Reverse Transcription Kits (Thermo Fisher Scientific). For RT-qPCR, the cDNAs were amplified using KAPA SYBR FAST qPCR Master Mix (KAPA Biosystems) and monitored using the LightCycler 480 System (Roche). For semi-quantitative RT-PCR, cDNAs were amplified by PCR to unsaturated levels, separated by electrophoresis, and stained with ethidium bromide. Images were obtained with a ChemiDoc system (BioRad) and analyzed with ImageJ software (NIH). Primers for PCR are shown in Supplementary Table 3.

**Cell viability assay**. Effect of small molecule treatment on cell viability was determined by WST-8 assay (CCK-8 kit, Dojindo). HeLa cells were seeded at $2 \times 10^4$ cells/well in 96-well plates and cultured for 24 h. Small molecules were added to final concentrations of 5 μM to the medium and the cells were cultured further for 24 h. CCK-8 solution (5 μl) was added to each well and the plates were further incubated for several hours at 37 °C before the measurement of the absorbance at 450 nm by a plate reader (EL808, BioTek).

**Fly experiments**. Instant *Drosophila* Medium Blue containing dry yeast was mixed with ultrapure water or 100 μM compound. Parent flies (*GMR-Gal4/+*; and *UAS-(UGGAA)*$_{exp}$/+ for *Drosophila* expressing r(UGGAA)$_{exp}$ or *GMR-Gal4/+*;; and *UAS-(UAGAA)(UAAAAUAGAA)*$_{exp}$/+ for *Drosophila* expressing r(UAGAA) (UAAAAUAGAA)$_{exp}$) were crossed on the food without and with 100 μM compound, and the offspring were generated on the same food at 25 °C. The eye morphology of the 1–2-day-old flies was analyzed using the stereoscopic microscope model SZX10 (Olympus).

**Reporting summary**. Further information on research design is available in the Nature Research Reporting Summary linked to this article.

## Data availability

Solution structure of the complex of naphthyridine carbamate dimer and an RNA with UGGAA-UGGAA pentad have been deposited with Protein Data Bank (https://www.rcsb.org) under accession number 6IZP. The data supporting the findings of this study are available from the corresponding author upon reasonable request. Source data are provided with this paper.

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

## Acknowledgements

This work was supported by Grant-in-Aid for Specially Promoted Research (26000007) and Scientific Research (A) (19H00924) for K.N. (Nakatani), for Scientific Research (C) (17K01962) and Scientific Research (B) (20H02880) for T.S., for Scientific Research on Innovative Areas (Brain Protein Aging and Dementia Control) (17H05699) for Y.N. from the Japan Society for the Promotion of Science (JSPS), by the Research Program of "Dynamic Alliance for Open Innovation Bridging Human, Environment and Materials" in "Network Joint Research Center for Materials and Devices" for K.N. (Nakatani), by Strategic Research Program for Brain Science (JP20dm0107061) and Practical Research Projects for Rare/Intractable Diseases (JP16ek019018, JP19ek0109222, JP20ek0109316)

for Y.N. from Japan Agency for Medical Research and Development, Japan, and by Intramural Research Grants for Neurologic and Psychiatric Disorders (30-3, 30-9) for Y.N. from the National Center of Neurology and Psychiatry.

## Author contributions

T.S. carried out screening and biophysical analysis, K.N. (Ninomiya) and T.H. carried out in vitro pulldown assay and nSBs experiments, G.K. and K.N. (Nagano) carried out structural analysis of RNA-NCD complex by NMR spectroscopy, T.S., M.U., Y.N. and K.I. carried out *Drosophila* experiments, T.S., Y.N., and K.N. (Nakatani) made a project plan, and T.S., G.K., T.H., and K.N. (Nakatani) wrote the manuscript.

## Competing interests

The authors declare no competing interests.
