## [Peer Review File · Nature Communications]

Editorial Note: This manuscript has been previously reviewed at another journal that is not operating a transparent peer review scheme. This document only contains reviewer comments and rebuttal letters for versions considered at Nature Communications .

Reviewer #1 (Remarks to the Author):

I was a previous reviewer of this manuscript. And, the authors have comprehensively responded to all questions. There is some debate as to the applicability of the models used in this study, especially fruit flies, to disease but all-in-all this is a solid contribution. I assume the other previous reviewers would be sent this paper and i am sure they will agree that the paper is now fine for publication. As usual, the nakatani group is quite professional and responsive to reviewer comments. we are appreciative of that and of the work in this manuscript.

Reviewer #2 (Remarks to the Author):

The manuscript by Shibata et al. reports the discovery and characterization of naphthyridine carbamate dimer (NCD) as a bioactive small molecule that targets disease-causing RNA repeat of r(UGGAA)_n. The reviewer reviewed the original manuscript when it was submitted to Nature Chemical Biology, and was very supportive of the work. With additional experiments and clarifications, the revised manuscript has addressed the reviewer's concerns for the original manuscript. Hence, the review recommends publication of this excellent study, which should be of significant interests to the broad readers of Nature Communications.

Reviewer #3 (Remarks to the Author):

The Authors have addressed most of my concerns and this is an improved manuscript. The work is of interest to the field and convincing, the conclusions are supported by the data. This is a good initial study showing the binding of NCD to toxic RNA repeats and the rescue of a fly model of disease. Repeat expansion disorders are extremely debilitating, degenerative conditions, with no cures are currently available. This study paves the way to future studies that will refine these molecules and assess relevance to human disease.

The work is appropriate for publication in Nature Communications.